# *ABCG2* Gene Expression in Non-Small Cell Lung Cancer

**DOI:** 10.3390/biomedicines12102394

**Published:** 2024-10-19

**Authors:** Agnieszka Jeleń, Marta Żebrowska-Nawrocka, Mariusz Łochowski, Dagmara Szmajda-Krygier, Ewa Balcerczak

**Affiliations:** 1Department of Pharmaceutical Biochemistry and Molecular Diagnostics, Medical University of Lodz, Muszynskiego 1, 90-151 Lodz, Poland; marta.zebrowska@umed.lodz.pl (M.Ż.-N.); dagmara.szmajda@umed.lodz.pl (D.S.-K.); ewa.balcerczak@umed.lodz.pl (E.B.); 2Laboratory of Molecular Diagnostics, BRaIn Laboratories, Medical University of Lodz, Czechoslowacka 4, 92-216 Lodz, Poland; 3Department of Thoracic Surgery, Copernicus Memorial Hospital, Medical University of Lodz, Pabianicka 62, 93-513 Lodz, Poland; mariusz.lochowski@umed.lodz.pl

**Keywords:** ABCG2, BCRP, cancer biomarker, DNA methylation, gene expression, LUAD, lung cancer, LUSC

## Abstract

**Background/Objectives:** ATP-binding cassette subfamily G member 2 [ABCG2/breast cancer resistance protein (BCRP)] contributes to mechanisms of multidrug resistance (MDR) and is a marker of side population (SP) cells in human cancers. The primary objective of this study was to investigate the impact of *ABCG2* gene expression on the non-small cell lung cancer (NSCLC) development, course of cancer disease, and patient prognosis using publicly available data. Obtained results were supplemented with assessment of *ABCG2* expression in blood of NSCLC patients. **Methods:** The dataset of lung cancer was analyzed utilizing the TIMER 2.0, UALCAN, TNMplot, MEXPRESS, cBioPortal, MethSurv, KM Plotter, STRING, and ShinyGO 0.80 databases. Blood samples from 50 patients were assessed using the real-time PCR method. **Results:** The *ABCG2* gene was expressed at a low level in NSCLC, and did not correlate with clinical aggressiveness of lung cancer. Higher *ABCG2* expression improved overall survival, but only in LUAD. In addition, CpG sites located on the CpG island affecting the NSCLC patient’s prognosis were indicated. In the case of our own laboratory results, the study did not reveal any changes in the *ABCG2* expression levels in blood collected from patients at different time points during the diagnostic–therapeutic procedure. In the in silico analysis, most ABCG2 protein interactors were associated with the development of drug resistance. **Conclusions:** ABCG2 appears to have a particularly significant impact on the survival of patients with lung cancer and on the effect of immunotherapy related to immune cell infiltration. Presented findings may support personalized medicine strategies based on bioinformatics findings.

## 1. Introduction

Lung cancer is the most common cancer among men and the second most common in women worldwide. This type of cancer is also the leading cause of cancer-related deaths among over 30 groups of neoplasms in half of the regions analyzed worldwide (among both, men, and women). While tobacco smoking remains the most important factor predisposing to the development of the disease [1,2], up to a quarter of all lung cancer cases occur in lifelong never-smokers. The incidence of lung cancer is decreasing as smoking rates decline; however, the incidence of lung cancer in never-smokers remains stable or is increasing, and may depend on gender, age, or ethnicity. In addition to the effects of smoking, increasing attention is being paid to the indoor/workplace exposure to radon, asbestos, arsenic, cooking oil, or exhaust fumes and level of other pollutants in the air we live in. Other related factors are family lung cancer history, diet, and infectious and inflammatory diseases [3,4].

NSCLC is the most commonly diagnosed histological type of lung cancer. The constant rate of non-smokers developing NSCLC highlights the important role of non-modifiable factors that are associated with carcinogenesis in lung tissue. Differences between smokers and non-smokers with NSCLC concern features such as gene expression patterns or germline polymorphisms [5,6,7]. Moreover, the major obstacles to the successful management of lung cancer are diagnoses at advanced or locally advanced stages, a limited number of markers to predict the response to applied therapy and monitoring its efficacy, and the development of resistance to therapy, leading to nonresponse or insufficient response. Therefore, in this study, we analyze a promising molecular factor that may be important for the development of NSCLC and bring benefits in clinical practice.

The members of the ATP-binding cassette (ABC) transporter superfamily are primarily known to significantly alter the pharmacokinetics of most anticancer drugs, including traditional chemotherapeutics, as well as molecularly targeted therapeutic agents. However, according to our previous study, the gene encoding the best-known ABC transporter—ABCB1—may also be a promising indicator of carcinogenesis and a disease prognostic factor, which could contribute to the alteration of the cancer progression and survival of patients [8]. The second factor of similar importance may be ABCG2 (ATP-binding cassette superfamily member G 2). The overexpression of ABCG2 associated with drug efflux may be among mechanisms of multidrug resistance (MDR), leading to failure in cancer therapy. The efficacy of the targeted inhibition of ABCG2-mediated transport has been investigated in various cancers such as colon cancer [9], breast cancer [10], and leukemia [11]. After the first disappointing results, it was necessary to continue the search for new strategies, promising multipotential MDR modulators and further preclinical tests. In lung cancer, an association was found between the ABCG2 transporter and the frequency of osimertinib adverse events [12], the risk of resistance to platinum therapy [13], or the adagrasib penetration of target tissues [14]. New MDR modulators such as tazemetostat [15], an I-CBP112-CBP/EP300 bromodomain inhibitor [16], or sonidegib [17] may increase the tumor cells’ sensitivity to anticancer agents by the functional inhibition of ABCG2. However, these new drugs could bring the most benefits in the clinical management of NSCLC patients with high ABCG2 expression.

Under normal physiological conditions, members of the ABCG subfamily are responsible for the regulation of cholesterol and bile acid homeostasis, steroid hormone transport, heme metabolism, and hypoxic signaling. Their presence at the strategic sites in the body along with the results of studies conducted on animals and in human cancer samples suggest that ABCG members play an important role in tumor formation and several other cancer-related processes. In addition, ABCG2 is a marker of the side population (SP) cells, representing pluripotent cancer stem cells. According to more recent data, the features of cancer that are significantly associated with activity of ABCG2 include genomic instability, resistance to cell death, plasticity of the cancer stem cell phenotype, sustaining proliferative signaling/ability to evade growth suppression, epigenetic reprogramming, and polymorphic differences in the microbiomes [18,19].

ABCG2 seems to be an important element of defense against carcinogens and factors that can stimulate inflammation. This was proven in a mouse model where the knockout of the gene analog caused toxic reactions and carcinogen accumulation [20,21]. ABCG2 also has the ability to pump out some of the polycyclic aromatic hydrocarbons contained in tobacco smoke. Therefore, it protects both normal lung cells and cancer cells from the effects of carcinogenic toxicity [22].

One of the biological effects of prolonged exposure to carcinogens and/or substances that affect the immune system is the production of reactive oxygen species (ROS), which may induce, e.g., genomic instability, epigenetic modifications, immunomodulation, or chronic inflammation, any of which can initiate carcinogenesis. The results obtained in a human colorectal adenocarcinoma cell line indicated that ROS promoted inflammation, which was more intense when *ABCG2* expression was decreased. It is likely that the down-regulation of the ABCG2 contributes to the activation of the NF-κB signaling pathway, which controls multiple aspects of the immune response [23]. At the same time, molecular changes in ABC transporters (including ABCG2) may have an effect on the transport of chemical compounds affecting the microbiome and/or produced by them [24]. Similarly to colorectal cancer, inflammation in the respiratory tract may be preceded by an altered lung microbial community composition and microbiome outside the lungs [25]. The diet, environmental factors, and ABC transporters mutually influence the complex network of interaction between the lung, oropharynx, and gut microbiome. For example, a significantly increased risk of lung cancer has also been observed in patients with lower respiratory tract infections or a *H. pylori* presence [26,27].

Cancer tissue is organized in hierarchical heterogeneous cell populations that exhibit distinct phenotypes and function within the tumor. Among them and of particular interest are cancer stem cells (CSCs), which are derived from the initiated normal stem/progenitor cells and have the ability for self-renewal and differentiation into multiple cell types. ABCG2 plays an important role in the enhancement in CSC tumorigenic potential. SP cells isolated from lung cancer cell lines demonstrated a higher level of ABCG2 at the mRNA and protein levels, providing them a broad spectrum of drug/xenobiotics resistance [28,29]. The expression of ABCG2, which is responsible for phenotypic characteristics of SP cells, is also involved in the WNT/β-Catenin, Hedgehog, Hippo, and PI3K/Akt/mTOR signaling pathways [30,31,32]. Therefore, ABCG2 status has a significant impact on tumor initiation and progression, the risk of local recurrence and metastasis to another part of the body, and patient survival.

This study involves the analysis of molecular and phenotypic datasets collected by The Cancer Genome Atlas (TCGA) and other bioinformatics databases. Using massive amounts of data from multiple studies and centers can yield more robust results and more reliable findings. Cancer bioinformatics plays an important role in the preliminary identification and/or authentication of promising biomarkers, which are revealed from individual clinical-case research. An in-depth study of *ABCG2* gene expression becomes possible with the selection and processing of transcriptomic and epigenomic data collected during multiomics studies in non-small cell lung cancer [33,34].

Here, we primarily investigate the impact of *ABCG2* gene expression on the NSCLC development, course of cancer disease, and patient prognosis using data collected in databases. Analyses conducted using bioinformatics tools are additionally supplemented with the results of our own wet research. In this study, we also explore the metabolic pathways in which ABCG2 is involved and its interactions with other proteins. The obtained results were compared with the available literature and discussed thoroughly.

## 2. Materials and Methods

### 2.1. Description of Data Collection Methods and Bioinformatic Tools

#### 2.1.1. A Pan-Cancer Analysis of the Expression of *ABCG2*

TIMER 2.0 (http://timer.cistrome.org, accessed on 4 May 2024) and TNMplot (https://tnmplot.com/analysis/, accessed on 4 May 2024) databases were used to analyze and present the differences in mRNA levels of *ABCG2* between normal and tumor tissue in different types of cancers. In the TIMER 2.0 tool, the Gene_DE module was chosen. This allowed for the identification of TCGA tumors in which the investigated gene was up- or down-regulated. The statistical significance computed by the Wilcoxon test was indicated by the number of asterisks displayed above the box plots [35,36]. The similar analysis in the TNMplot tool included not only samples from TCGA but also data from Gene Expression Omnibus (GEO), The Genotype-Tissue Expression Project (Gtex), and Therapeutically Applicable Research to Generate Effective Treatments (TARGET) databases. The differences in *ABCG2* expression were determined using the Mann–Whitney U test. The assessment of differences in the *ABCG2* mRNA level between lung cancer tissue (LUAD and LUSC) and adjacent normal tissues were performed using RNA-Seq data [37].

#### 2.1.2. Association between *ABCG2* Expression and Clinicopathological Characteristics

The expression levels of *ABCG2* in lung cancer tissues were further analyzed using two tools: UALCAN (https://ualcan.path.uab.edu, accessed on 18 May 2024) and MEXPRESS (https://mexpress.be, accessed on 20 May 2024). The aim of this step was to indicate the potential association between the investigated gene expression level and clinicopathological as well as demographic features of lung cancer patients. The analyzed features included, e.g., the cancer stage; pathologic tumor, node, and metastasis (TNM) status; age; gender; ethnicity; mutation status in selected genes, and tobacco smoke exposure characteristics. In the UALCAN tool, the statistical significance of differences in *ABCG2* expression between lung cancer patients grouped according to selected feature was determined using Welch’s *t*-test [38,39]. For the same purpose, the MEXPRESS tool was used, in which the gene expression was analyzed based on more detailed clinicopathological data from TCGA. Data were sorted according to the *ABCG2* expression level, and then the appropriate statistical test was performed (ANOVA, t test, or Pearson correlation coefficient calculation). All presented *p* values are Benjamini–Hochberg-adjusted *p* values [40,41].

#### 2.1.3. *ABCG2* DNA Methylation Analysis

Using both UALCAN and MEXPRESS, a preliminary comparison of the *ABCG2* promoter methylation level (sum of data from CpG probes located up to 1500 bp upstream of gene’s start site) between cancer and normal lung cells was made, and correlation between the expression of this gene and DNA methylation data in relation to its genomic location was estimated. The association between the *ABCG2* mRNA expression level and methylation beta-value (HM27 and HM450 merge) was analyzed using the cBioPortal platform (https://www.cbioportal.org, accessed on 2 September 2024). A total of 508 LUAD patients/samples and 483 LUSC patients/samples from the TCGA PanCancer Atlas with complete RNA-seq data were included in the analysis. Spearman’s and Pearson’s correlation coefficients were calculated [42]. The MethSurv tool (https://biit.cs.ut.ee/methsurv/, accessed on 25 July 2024) was used to characterize the DNA methylation level of all CpG islands located in *ABCG2* LUAD- and LUSC-TCGA samples [43].

#### 2.1.4. Survival Analysis of *ABCG2*

Two survival databases, the Kaplan–Meier plotter (https://kmplot.com/analysis/, accessed on 30 May 2024) and above-mentioned MethSurv, were used to analyze the effect of *ABCG2* expression and methylation on patients’ prognosis. The gene expression data obtained by gene chips from The European Genome-phenome Archive (EGA), GEO, and TCGA databases were analyzed. Overall survival (OS), first-progression survival (FP), and post-progression survival (PPS) were compared between two groups of lung cancer patients (high vs. low expression level) split by the “auto select best cutoff” option. The selected follow-up threshold was 60 months, and the biased arrays were excluded for array quality control. Similar analyses were performed in subgroups restricted to the histological type of lung cancer, cancer stage, tumor grade, chemotherapy reception, or smoking status. For each analysis, the hazard ratio (HR) with 95% confidence intervals (CIs) and *p* value by the log-rank test were calculated [44]. A survival analysis based on DNA methylation levels for any CpG site was performed using the MethSurv tool. The output visualized by the Kaplan–Meier plot presents the differences in survival between groups of patients with higher and lower methylation levels split by the “best” option. The *p* value of HR with 95% CI, derived from Cox fitting and the log-likelihood ratio (LR) test, was calculated for each analysis [43].

#### 2.1.5. Immune Infiltration Analysis

We used the Gene module in the TIMER2.0 database to correlate *ABCG2* expression with the immune infiltration level in lung cancer tissue. B cells, CD8+T cells, CD4+T cells, macrophages, neutrophils, and dendritic cells were assessed by the TIMER algorithm. Cancer-associated fibroblasts were evaluated by MCP-COUNTER, XCELL, and TIDE algorithms. The “Purity adjusted” option was selected for all analyses. Spearman’s RHO coefficient and statistical significance value were presented on scatter plots. *p*  <  0.05 was considered as significantly different [35,36].

#### 2.1.6. Gene Mutation and Copy Number Alteration Analyses

The *ABCG2* mutations and their location were detected using cBioPortal (http://www.cbioportal.org/; accessed on 10 September 2024) [42,45,46]. The web tool enabled generating the graphical visualization of mutation and copy number alterations in selected cancer via Oncoprint and MutationMapper modules. The chosen dataset for graphical visualization was the TCGA and PanCancer Atlas, with selected mutations, Structural Variant, and putative copy number alterations from GISTIC and mRNA Expression Genomic Profiles.

#### 2.1.7. Analysis of ABCG2-Related Proteins and Gene Set Enrichment Analysis

To predict potential protein–protein functional interaction related to ABCG2, the STRING tool was used (https://string-db.org/; accessed on 7 September 2024) [47]. In the analysis, 20 interactions with a minimum confidence at level 0.7 and PPI enrichment *p* value of *p* < 9.26 × 10^−9^ were predicted. The k-means clustering of the generated PPI networks for ABCG2 was performed with a preset of three clusters (marked as red, green, and blue). A gene enrichment analysis was undertaken using the ShinyGO 0.80 tool (http:// bioinformatics.sdstate.edu/go/; accessed on 7 September 2024), in order to identify the most important functions of particular clusters of correlated genes [48].

### 2.2. Sample Collection

The material was peripheral blood collected from patients hospitalized in the Department of Thoracic Surgery, Memorial Copernicus Hospital, in Lodz, and diagnosed with NSCLC. Blood samples were collected in the years 2022–2023. The study was conducted in accordance with the principles of the Declaration of Helsinki, and all patients provided informed consent prior to enrollment. The research protocol was approved by the Ethical Committee of the Medical University of Lodz (no. RNN/85/20/KE and no. KE/649/23).

This study involved 50 patients (13 females; 37 males; median age of the group: 67 years), of which 1 patient was excluded before the publication of results. Blood samples were collected before surgical removal of lung tumors, in all patients. In the case of 6 study participants, blood was also collected 100 days and one year after the surgical removal of the tumor. Basic clinical data (gender, age, smoking status, histological type of cancer, TNM stage, histological grade, and type of treatment, if implemented) were obtained from each patient. Detailed characteristics of the study group are presented in Table 1.

#### 2.2.1. RNA Isolation

Total RNA from peripheral blood was isolated according to the “Total RNA Mini” protocol (A&A Biotechnology, Gdansk, Poland). The purity and concentration of RNA samples were assessed spectrophotometrically. RNA samples until analyses were stored at −80 °C.

#### 2.2.2. Real-Time Polymerase Chain Reaction (RT-qPCR)

Isolated RNA was transcribed into complementary DNA (cDNA) in accordance with the “High-Capacity cDNA Reverse Transcription Kit” with the RNase Inhibitor protocol (Applied Biosystems by Thermo Fisher Scientific Baltics, UAB, Vilnius, Lithuania). The final concentration of RNA in the reaction mixture was 0.005 µg/µL. After that, quantification assessment of *ABCG2* mRNA was performed by the real-time PCR method using the Rotor-GeneTM6000 (Corbet Research, Qiagen GmbH, Hilden, Germany) according to the “iTaq Universal SYBR Green Supermix” (Bio-Rad, Hercules, CA, USA) protocol. *GAPDH* was used as the reference gene. The reaction mixture for both investigated and reference genes consisted of 5 µL of 2x concentrated master mix, 0.5 µL of each primer (10 mM) (*ABCG2* gene: F 5’-ATG TCA ACT CCT CCT TCT AC-3’, 5’-AAT GAT CTG AGC TAT AGA GGC-3’; *GAPDH* gene: 5’-ACA GTT GCC ATG TAG ACC-3’, R 5’-TTG AGC ACA GGG TAC TTT A-3’), and 1 µL of cDNA or distilled water for a negative control (NTC) up to a final volume of 10 µL. All testing was performed in technical triplicates. The final Ct value for each sample was calculated as a mean of 3 replicates. In each experiment, NTC was included. To define the level of relative *ABCG2* gene expression, the ΔΔ Ct method was used.

#### 2.2.3. Statistical Analysis

To evaluate the correlation between the relative levels of investigated gene expression and age of patients, Spearman’s rank correlation coefficient was calculated. To identify the potential differences in *ABCG2* expression between the lung cancer patients grouped according to clinicopathological features, the U Mann–Whitney test was used. Friedman’s ANOVA was applied to compare the median of the *ABCG2* expression between the patients “before surgery”, “100 days after surgery”, and “one year after surgery”. All statistical analyses were performed using STATISTICA 13.1. (StatSoft Inc., Tulsa, OK, USA). In all conducted analyses, a *p* < 0.05 was assumed as significant.

## 3. Results

### 3.1. ABCG2 Gene Expression Pattern

The results of the pan-cancer analysis demonstrated that the level of *ABCG2* gene expression differed significantly between tumor and normal tissue in more than half of the cancer types (16 out of 23). For 15 types of cancer, including lung cancer, a reduction in expression was observed in tumor tissue. Kidney renal cell carcinoma (KIRC) was an exception (Figure 1A). When samples deposited in databases other than TCGA were included in the analysis, there were no significant changes in outcomes (Figure 1B). Similarly, *ABCG2* expression was down-regulated in both LUAD (Figure 1C) and LUSC (Figure 1D) compared with adjacent normal tissue.

Furthermore, the associations of *ABCG2* expression with clinicopathological and demographic features in lung cancer were evaluated. The investigated gene expression was significantly higher in LUSC stage 4 than 3 (*p* < 0.05; Appendix A). For LUAD, the pathologic N stage was correlated with *ABCG2* expression (*p* = 0.0190; Appendix A). Additionally, this gene expression was higher in Caucasians than African Americans, for both LUAD (*p* = 0.0150, Appendix A) and LUSC (*p* = 0.0480, Appendix A). No associations were observed between the *ABCG2* mRNA level and other characteristics, including cigarette smoking, *EGFR*, *KRAS* and *TP53* mutation status, pathologic T stage, and M stage.

### 3.2. Hypomethylated ABCG2 Gene in Lung Cancer

To explore the reasons for the down-regulation of *ABCG2* expression in lung cancer, the methylation level of the investigated gene was analyzed in three ways. As shown in Figure 2A and Figure 3A, in both LUAD and LUSC samples, the hypomethylated *ABCG2* promoter region was significantly less methylated in cancer samples than normal samples (*p* < 0.001). As is known, DNA methylation is a crucial epigenetic mark that influences gene expression. However, neither in LUAD (Figure 2B,C) nor in LUSC (Figure 3B,C) was there an association between the *ABCG2* gene expression and DNA methylation level in the analyzed regions. Furthermore, we found that cg02016771 was the most methylated (hypermethylated) site in this gene, in both LUAD (Figure 2D) and LUSC (Figure 3D). Most regions of the *ABCG2* gene remain hypomethylated. It is worth noting that an abnormal DNA methylation pattern as one of the hallmarks of cancer may correlate with the patient prognosis.

### 3.3. Prognostic Value of ABCG2 Gene Expression and Methylation

To evaluate the prognostic significance of *ABCG2* gene expression in patients with lung cancer, Kaplan–Meier curves for OS, FP, and PPS were generated and analyzed. A higher expression of *ABCG2* in LUAD patients was associated with improved OS (*p* = 0.0190, Figure 4B) and FP (*p* = 0.0034, Figure 4E). In contrast, a higher *ABCG2* mRNA level was significantly associated with poorer FP (*p* = 0.0200, Figure 4F) and PPS (*p* = 0.0130, Figure 4I) in the LUSC group. It is worth noting that the number of participants involved in the analysis of PPS in LUAD was relatively small. Therefore, there is most likely no significant correlation between the expression of the investigated gene and PPS in lung cancer. The *ABCG2* gene expression level appears to have the greatest impact on the prognosis of LUAD patients (Figure 4).

Next, all LUAD cases were divided into two or three subgroups based on tumors’ clinical/pathologic features and patient demographics. The OS analysis was performed for each of these subgroups (Appendix A). In this type of cancer, improved OS was correlated with high *ABCG2* expression only in early-stage cancer disease (stage 1) (*p* < 0.001, Appendix A), in high-grade tumors (G3) (*p* = 0.0.340, Appendix A), in patients who never smoked (*p* = 0.0220, Appendix A), and regardless of gender (*p* = 0.0400, *p* = 0.0021, Appendix A). In contrast, higher *ABCG2* expression was significantly associated with poorer OS in patients treated with chemotherapy (*p* = 0.0270, Appendix A).

Finally, we explored the prognostic value of DNA methylation levels of the *ABCG2* gene. The status of the only hypermethylated region (cg02016771, 5’UTR, Open_Sea) had no effect on OS of lung cancer patients (Figure 2E and Figure 3E). Of the most poorly methylated CpG sites, the higher methylated cg03415858_TSS200, cg02196227_TSS1500, or cg27493371_TSS1500 slightly improved LUAD patients’ survival. However, these observations were not statistically significant (Figure 2G–I). In the case of LUSC, CpG sites located on the CpG island, which influence the patient’s prognosis, were successfully selected. Low methylated cg01263075_TSS200 has better survival statistics (*p* = 0.038, Figure 3F). On the other hand, a higher methylation of two CpG sites (cg03415858 and cg25295218) located up to 200 bp upstream from the transcription start site in the *ABCG2* gene was associated with better OS (*p* = 0.026, Figure 3G, and *p* = 0.023, Figure 3H, respectively). This suggests that unmethylated *ABCG2* takes part in lung squamous cell carcinoma progression.

### 3.4. The Associations between Immune Infiltrates and ABCG2 Gene Expression

The expression of the ABCG2 transporter can be regulated by diverse signaling molecules derived from immune cells. On the other hand, cytokines are transported and controlled by various ABC family proteins [49,50]. This indicates a link between the presence of immune infiltration cells and the function of transporters. The close correlation between the expression of *ABCG2* and macrophages, neutrophils, dendritic cells, and CD8+ T cells was found in LUAD based on the TIMER algorithm (Figure 5A). In the LUSC subtype, *ABCG2* expression showed positive correlations with the same immune cell types, except neutrophils (Figure 5B). Furthermore, the level of infiltration with cancer-associated fibroblasts was a distinguishing feature of the investigated histological subtypes of lung cancer. Based on the MCP-COUNTER, XCELL, and TIDE algorithms, a significant positive correlation between *ABCG2* expression and cancer-associated fibroblast infiltration was observed in LUAD, but not in LUSC (Figure 5C,D).

### 3.5. ABCG2 Expression in the Blood of Lung Cancer Patients Based on Our Wet Analysis

After analyzing the data available in online databases, we decided to check whether similar associations could be identified in the blood of patients with lung cancer. In the study group, the differences in relative expression of *ABCG2* in the blood samples were observed (median: 1.01; min–max: 0.03–6.92). Next, it was assessed whether there was a correlation between the clinicopathological features of investigated subjects and the level of *ABCG2* gene expression in peripheral blood. There was no association between *ABCG2* mRNA in blood of NSCLC patients and the age at diagnosis (*p* = 0.1995, Appendix A), histological type of the cancer (*p* = 0.2638, Appendix A), grade of histological malignancy (*p* = 0.6334, Appendix A), cancer stage (*p* = 0.8486, Appendix A), gender (*p* = 0.8433, Appendix A), and cigarette smoking (*p* = 0.3604, Appendix A). Additionally, we checked whether the expression of the investigated gene changes after tumor removal/as a result of the applied therapy. However, there were no differences in the *ABCG2* gene expression between the groups of patients before surgery and 100 days and one year after the surgery (*p* = 0.8187). Since it is the first such investigation, performed in a restricted study group, the unequivocal exclusion of the surgical tumor removal effect (and/or chemotherapy) on gene expression in patients with lung cancer is impossible. Data are summarized in Appendix A.

### 3.6. Analysis of ABCG2 Gene Mutations and Copy Number Variations in LUAD and LUSC

*ABCG2* gene alterations were studied using the cBioPortal database utilizing the TCGA PanCancer Atlas set. Of the total samples queried (503 for LUAD and 466 for LUSC), the *ABCG2* gene was altered in 2% and 3%, respectively (Figure 6A and Figure 7A), while the somatic mutation frequency was 2.4% for both subtypes (Figure 6B and Figure 7B). In both, the missense variants dominate (10 in LUAD and 8 in LUSC). Then, the association between the *ABCG2* expression level and mutation as well as copy number alteration status was evaluated for LUAD (Figure 6C,D) and LUSC (Figure 7C,D). No significant differences were found, apart from lower levels of mRNA expression z-scores compared to normal tissue in the case of LUSC samples with deep deletion (*p* < 0.01).

### 3.7. ABCG2 Functional Enrichment and Pathway Analysis

The in silico analysis showed the protein–protein interaction of the ABCG2-centered network. According to obtained results, ABCG2 functional partners, with a confidence score of at least 0.7, were ABCC2; ABCC1; SLCO1B1; SLC2A9; SLC22A12; BSG; MRPS7; SLC22A; SLC22A8; ALB; SLC17A1; CYP3A4; SLCO1B3; PROM1; SLCO2B1; SLCO1A2; NR1I2; SLC47A1; SLC22A11; and ERBB2. See Figure 8.

Within the generated PPI network, three clusters of interaction proteins were identified: red, green, and blue. In the red and green clusters, interactions indicate that members of these clusters are mostly transporters, for example, of organic cations (SLC22A1) or the excretion/detoxification of endogenous and exogenous organic anions (SLC22A1), phosphate into cells via Na^+^ cotransport, urate transport in the kidney (SLC17A1), etc. (Figure 8A). In the case of a blue cluster, it consists of three proteins involved in the metabolism of sterols, steroid hormones, retinoids, and fatty acids (CYP3A4); activation of transcription factors, involved in the metabolism and secretion of xenobiotics, drugs, and endogenous compounds (NR1I2); mediation of the Na^+^-independent uptake of organic anions; and clearance of bile acids and organic anions from the liver (SLCO1B1). The analysis of gene enrichment revealed the most important pathways modulated by the *ABCG2* gene and its product. A notable enrichment of gene clusters composed primarily of transporters (red and green) in bile secretion, transmembrane transport via ATP-dependent proteins, and platinum drug resistance pathways was identified (Figure 8B). In addition, the blue cluster contained genes associated with linoleic acid and retinol metabolism, steroid hormone biosynthesis, chemical carcinogenesis, and the metabolism of drugs and xenobiotics via cytochrome P450 and other enzymes (Figure 8C).

## 4. Discussion

ABCG2 is highly expressed in placental syncytiotrophoblasts, the cerebral endothelium, and epithelial cells in the intestine and kidney [51] but can also be found in cells localized in lung tissue. A small amount of this protein is identified in the epithelial layer, in seromucinous glands, and in small endothelial capillaries of lung tissue. Simultaneously, high ABCG2 levels were found in many cell lines of a pulmonary origin [52]. The unambiguous identification of lung cells that express this protein remains difficult and controversial because the up-regulation of ABCG2 is mainly drug-induced, resulting in a multidrug-resistant cell phenotype Therefore, in some types of cancer, e.g., myeloid leukemia [51] or lung cancer [53], a high expression of the ABCG2 is associated with a negative patient prognosis. However, this transporter has particular relevance in the lung, where its activity may be critical not only for drug distribution but also in providing protection against inhaled toxins, and other xenobiotics.

*ABCG2* is expressed at a low level in most TCGA cancers. This indicates an activation of common molecular mechanisms that are responsible for reduction in cellular *ABCG2* expression, which is associated with the ongoing process of carcinogenesis [54]. However, the mechanisms by which the *ABCG2* gene becomes underexpressed in cancers remain unknown, as well as relatively little being known about the mechanisms that physiologically regulate the expression of this gene. A possible mechanism might be DNA methylation, regulation through miRNA, or cytokine and growth factor activation in the lung cell environment. On the other hand, ABCG2 expression is high in SP cells isolated from lung cancer cell lines and other tumor cell lines. This subtype of cells has been identified as promoting carcinogenesis. However, the fraction of these cells is so small that it should not significantly influence the results of the assessment of the *ABCG2* mRNA level in tumor tissue.

*ABCG2* expression levels were shown to be significantly lower in lung tumors than in normal tissues. It is consistent with other findings in primary tumor tissues from LUAD patients [55]. A significant correlation between pathological cancer stages and *ABCG2* expression was shown; its expression was higher in stage 4 in comparison with stage 3 of LUSC. However, the group of patients in the last stage of lung cancer was small, which could have influenced the obtained result. Overall, the *ABCG2* gene expression level did not correlate significantly with the age at the time of diagnosis, pathological cancer stage, cigarette smoking, ethnicity, or gender of lung cancer patients. Therefore, *ABCG2* expression most likely does not affect the processes related to clinical aggressiveness of the disease in this type of cancer.

The therapeutic approach in NSCLC depends on the presence or absence of specific mutations, which in this type of cancer occur most frequently in the *EGFR* or *KRAS* gene. Since LUAD and LUSC are characterized by different mutations, we hypothesized that the expression of the investigated gene would correlate with driver mutations’ status of *EGFR* or *KRAS*. However, the results obtained from analyses in the MEXPRESS and UALCAN web tools did not confirm this hypothesis. It was shown that the mutations in one of three genes (*EGFR*, *KRAS*, and *TP53*) did not significantly change the *ABCG2* expression level, while some associations were observed in the study by Jaromi et al. Generally, in the *KRAS* mutant cell line, *ABCG2* expression was higher than in the *EGFR* mutant cell line. Furthermore, it was dependent on the expression of genes from the Wnt signaling pathway [55]. Nevertheless, it should be emphasized that the expression profile of the *ABCG2* in primary tumor tissues differs significantly from that observed in cell cultures [52]. It seems that cellular processes and strings of signaling pathways regulating activity of ABC transporters are complex and strongly dependent on environmental conditions; therefore, the best material for research may be primary patient samples, not subjected to prior manipulations or passages.

Next, we divided the lung cancer patients according to the histological subtype and investigated the correlation of *ABCG2* expression with the prognosis of patients within LUAD, LUSC, and both subtypes combined. The overexpressed *ABCG2* gene showed a protective effect, but only in LUAD. This result is in contradiction with a lot of reports indicating a poor prognosis of patients whose cancer cells are rich in the ABCG2 protein [53,56,57,58]. However, the surface ABC protein levels do not always correlate with its transcript levels, suggesting that the mechanism responsible for control of ABCG2 concentration may affect the amount of protein, without changing the mRNA level [59]. Additionally, according to Liang et al., the cellular localization of the ABCG2 protein also includes the mitochondrial membrane and inside the cell nucleus. They found that nuclear ABCG2 may serve as a transcription regulator for the factors related to epithelial cell adherence and their polarization and differentiation [60]. We hypothesized that the ability of ABCG2 to shuttle between the cytoplasm and nucleus, or interaction between this protein and DNA, may be modified by the presence/level of the *ABCG2* transcript. Further analyses would be necessary to clarify the mechanism of interaction between ABCG2 protein and its transcript.

Promoter methylation is one of the important mechanisms for the regulation of gene expression. So far, the knowledge about the mechanisms of the transcriptional regulation of *ABCG2* expression remains limited in most types of human cancers. In lung cancer, knowledge on this subject is extremely poor. In this study, a low level of *ABCG2* gene promoter methylation was found in lung cells. In addition, in both LUAD and LUSC, the hypomethylated *ABCG2* promoter region was significantly less methylated in cancer samples than normal samples. The hypomethylation of *ABCG2* was also confirmed in most cases of gallbladder cancer [61] or breast cancer [62]. Additionally, the hypomethylation of this gene may show significant correlation with its increased expression [61]. Methylation status may also be significantly associated with response to chemotherapy [62]. In our analysis, we did not show any correlation between the *ABCG2* gene expression and DNA methylation level in the promoter region. In this regard, lung cancer appears to differ from gallbladder cancer. Moreover, according to our best knowledge, this study is the first to explore the prognostic value of DNA methylation levels of this gene. CpG sites located on the CpG island, which influence the patients’ prognosis, were selected, and it included cg01263075, cg03415858, and cg25295218 located up to 200 bp upstream from the transcription start site in *ABCG2*. Thus, an altered methylation of the defined gene region may be one of the important prognostic markers in lung cancer.

Cancer-associated fibroblasts are the dominant cell population in the tumor microenvironment. These cells promote tumorigenesis by initiating the extracellular matrix remodeling and secreting multiple growth factors and cytokines [63]. A positive correlation between cancer-associated fibroblast infiltration and *ABCG2* expression was observed in LUAD, but not in LUSC. Therefore, a higher expression of *ABCG2* could have a potential role in the prediction of tumorigenesis process activation in lung cells, especially in LUAD patients. Further analyses demonstrated a significant positive correlation between *ABCG2* expression and levels of multiple immune cells, which is consistent with the results of other research [64]. In summary, *ABCG2* is strongly involved in immunoinfiltration in lung cancer and therefore remains a promising target for immunotherapy related to immune cell infiltration.

The infiltration of immune cells, particularly macrophages, may be the result of EPHB2 overexpression, which is characteristic for LUAD. Eph receptor B2 (EphB2), an important member of the ephrin receptor family, can stimulate ABCG2 expression in other types of cancer cells, which can explain the correlation observed in our study [65]. In the case of other members of this superfamily, their overexpression has also been observed during the development of lung cancer. An example is EphB4, which has been linked to tumor angiogenesis, growth, and metastasis [66]. Assessment of the interaction between Eph receptors and ABCG2 requires further research.

In our study, the expression of the *ABCG2* gene was additionally evaluated in blood samples taken from patients at three times points (at NSCLC diagnosis and 100 days and one year after surgery). Unfortunately, there were no differences in the *ABCG2* gene expression between these time points. This means that the evaluation of *ABCG2* mRNA in the peripheral blood cells is most likely not suitable for assessing the effectiveness of therapeutic interventions in lung cancer patients. Due to the small number of analyzed samples, this statement should be verified in future research. It is the first such investigation, and thus the unequivocal exclusion of correlation between the *ABCG2* expression in blood and general effectiveness of anticancer treatment in patients with lung cancer is impossible.

The exploration of gene alterations can help to elucidate the gene function and role in lung cancer development and progression. cBioPortal was used to examine the frequency as well as the alteration layout across the *ABCG2* gene. The alteration frequency was rather low across LUSC and LUAD samples (less than or close to 3%), with missense variant domination. Overall, the observed mutations and copy number alterations did not have influence on the mRNA expression z-scores relative to normal samples. In their study, Wang et al. suggested that the mutation pattern of the *ABCG2* gene can be an independent risk factor for the NSCLC prognosis [67], although it remains unclear how this can influence gene interactions, and which mutation is characteristic for the chosen cancer type.

As indicated by the analysis using the STRING tool, the ABCG2 protein interacts mainly with other transporters. Available publications connected with this issue indicate that this protein participates in the ADME mechanisms for drugs. This is especially in the case of the cooccurrence of various polymorphisms for genes encoding proteins from the predicted PPI network. A connection has been shown between the occurrence of variants of the *ABCG2* and *NR1I2* genes and the pharmacokinetics of dolutegravir, used in HIV infection [68]. On the other hand, variants of the genes encoding SLCO1B1 and ABCG2 may affect the level of statins in the blood of patients treated with them [69,70]. Therefore, most of these interactions are associated with the potential development of drug resistance. For example, increased expressions of ABCC1 and ABCG2 reduced the therapeutic effect in an ovarian cancer cell line [71].

The *ABCG2* gene may also be involved in the process of carcinogenesis indirectly, by influencing such processes as apoptosis, autophagy, or ferroptosis. In a normal placenta, reduced or no expression of this gene results in increased sensitivity of placental trophoblast cells to apoptosis in response to cytokines or ceramides. Therefore, the presence of *ABCG2* gene expression protects the cell from apoptosis. However, in the case of cancer cells, this may have a negative effect, leading to an excessive proliferation of these cells and promoting carcinogenesis. Additionally, it has been shown that the overexpression of *ABCG2* in cancer cells enhances the process of autophagy induced by non-substrate stressors (such as radiation or nutrient starvation), contributing to increased cell survival and promoting cancer progression [72,73]. The ABCG2 as a potential heme exporter may be associated with the effect on ferroptosis, although the mechanism has not been fully elucidated, but it is believed that the amount of heme in the cell affects their sensitivity to this phenomenon. Reducing the amount of heme potentially lowers ferroptosis [74].

Although it has been shown in a mouse model that vascular endothelial progenitor cells with the overexpression of *ABCG2* have a higher potential for vessel formation in vivo compared to mature endothelial cells, no association has been demonstrated between *ABCG2* gene expression and *VEGF* gene expression and tumor vascularization in patients with retinoblastoma [75,76].

## 5. Conclusions

This study demonstrates the potential of *ABCG2* gene expression as well as its methylation evaluation as a biomarker in cancer through in silico and wet analyses. The low *ABCG2* mRNA level is a feature of NSCLC, with no correlation to clinical aggressiveness. This gene appears to have a particularly significant impact on the survival of patients because higher expression improved overall survival in LUAD patients. Additionally, the study highlighted CpG sites influencing the prognosis and drug resistance. The effect of immunotherapy related to immune cell infiltration also significantly depends on the *ABCG2* expression. To the best of our knowledge, this study is the first to present the most comprehensive analysis of the expression of this gene in NSCLC. Despite the fact that there were no differences of *ABCG2* gene expression in blood between subgroups of investigated lung cancer patients (before surgery and 100 days and one year after the surgical removal of the tumor), it is the first such investigation; hence, it was not possible to confront/discuss obtained results and draw unequivocal conclusions. Therefore, these findings require confirmation on a larger group of participants.

## Figures and Tables

**Figure 1 biomedicines-12-02394-f001:**
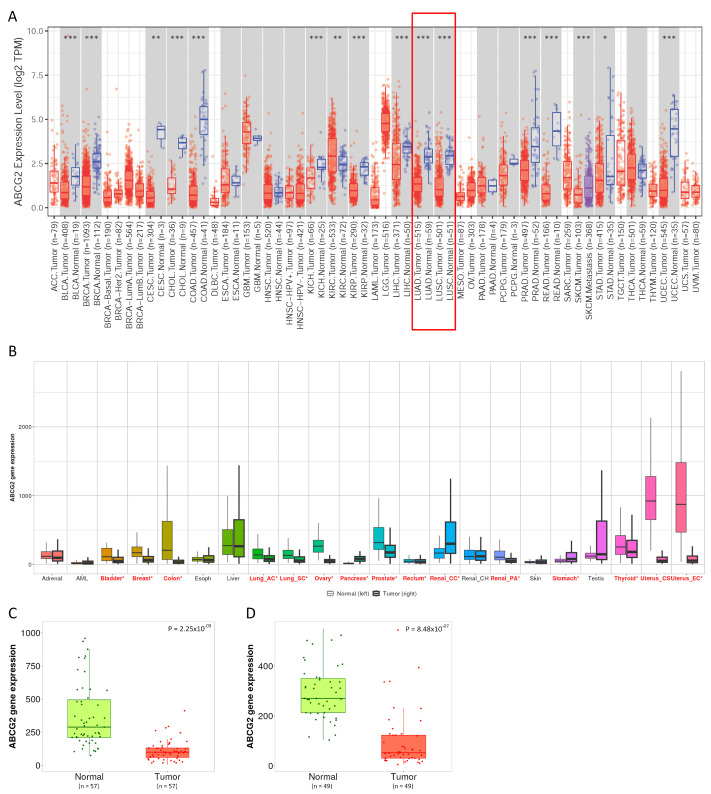
*ABCG2* expression in human pan-cancer and lung cancers. (**A**,**B**) The mRNA level in a panel of tumors (**A**) from the TIMER2.0 database (*p* value codes: 0 ≤ *** < 0.001 ≤ ** < 0.01 ≤ * < 0.05); the red box mark the box plots for lung cancer, which is the focus of this study (**B**) and TNMplot database (significant differences in expression between cancer and normal tissue are indicated by a red asterisk; *p* < 0.01); a red * marks those tissues for which the differences in the expression levels were statistically significant. (**C**,**D**) The down-regulation of *ABCG2* in (**C**) LUAD and (**D**) LUSC in comparison with adjacent normal tissue (RNA-seq data, TNMplot).

**Figure 2 biomedicines-12-02394-f002:**
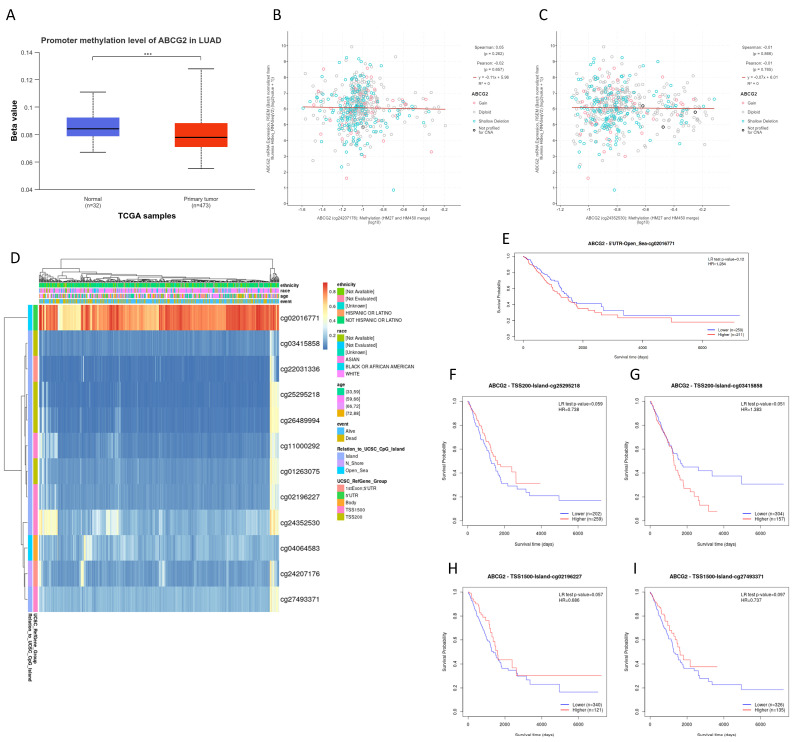
The DNA methylation analysis of the *ABCG2* gene in LUAD. (**A**) Comparison between the promoter methylation level between LUAD and normal lung tissue (*** *p* < 0.001; UALCAN). Correlation between the gene expression level and methylation of two *ABCG2* regions in LUAD samples ((**B**) cg24207176 and (**C**) cg24352530 probes; cBioPortal). (**D**) The heat map showing twelve methylated CpG sites of *ABCG2* (1 = fully methylated/red color; 0 = fully unmethylated/blue color; MethSurv). (**E**–**I**) Survival curves based on selected CpG methylation sites (MethSurv).

**Figure 3 biomedicines-12-02394-f003:**
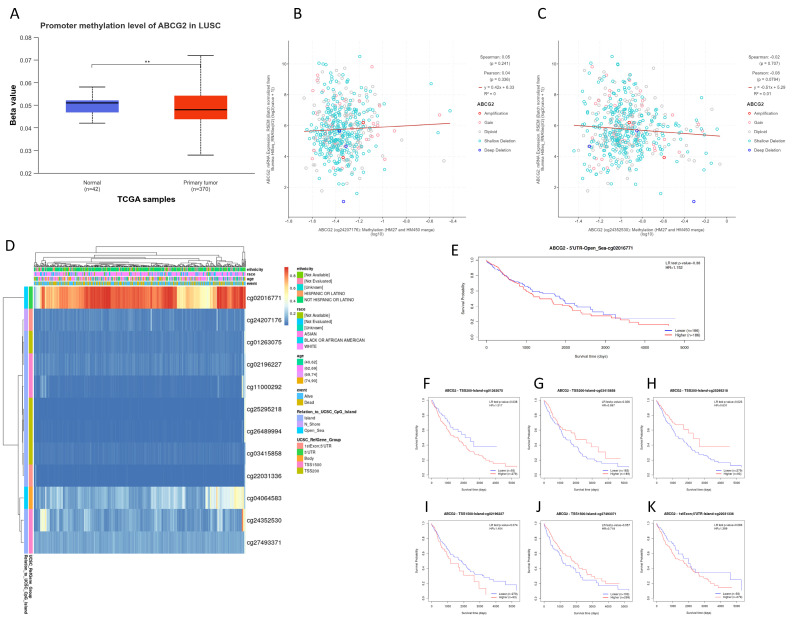
The DNA methylation analysis of the *ABCG2* gene in LUSC. (**A**) Comparison between the promoter methylation level between LUSC and normal lung tissue (** *p* < 0.01; UALCAN). Correlation between the gene expression level and methylation of two *ABCG2* regions in LUSC samples ((**B**) cg24207176 and (**C**) cg24352530 probes; cBioPortal). (**D**) The heat map showing twelve methylated CpG sites of *ABCG2* (1 = fully methylated/red color; 0 = fully unmethylated/blue color; MethSurv). (**E**–**K**) Survival curves based on selected CpG methylation sites (MethSurv).

**Figure 4 biomedicines-12-02394-f004:**
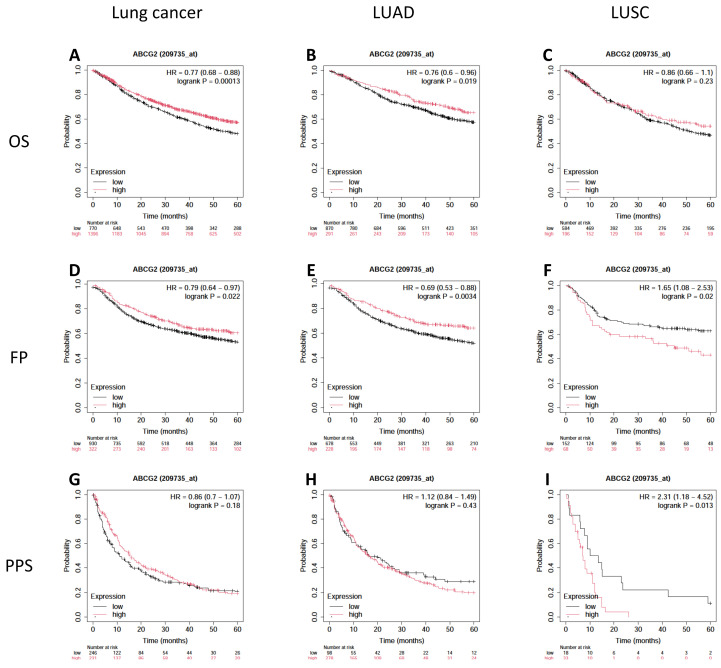
Kaplan–Meier survival curves of *ABCG2* (low vs. high expression level) overall (**A**,**D**,**G**) in non-small cell lung cancer, (**B**,**E**,**H**) in LUAD, and (**C**,**F**,**I**) in LUSC (*p* < 0.05; Kaplan–Meier plotter). OS, overall survival; FP, first-progression survival; PPS, post-progression survival; HR, hazard ratio; red: high expression; black: low expression.

**Figure 5 biomedicines-12-02394-f005:**
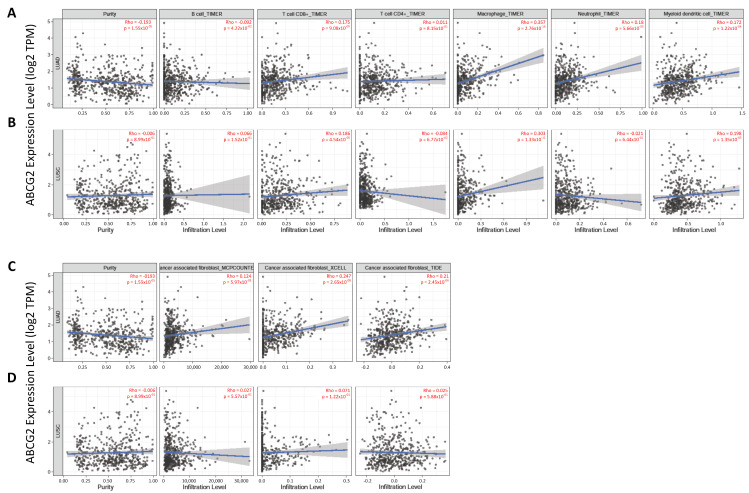
The association between *ABCG2* and tumor immune infiltration (B cells, CD8+ and CD4+ T cells, macrophages, neutrophils, myeloid dendritic cells) in (**A**) LUAD and (**B**) LUSC based on the TIMER algorithm. The correlation of the cancer-associated fibroblast infiltration level with (**C**) LUAD and (**D**) LUSC based on MCP-COUNTER, XCELL, and TIDE algorithms (TIMER2.0).

**Figure 6 biomedicines-12-02394-f006:**
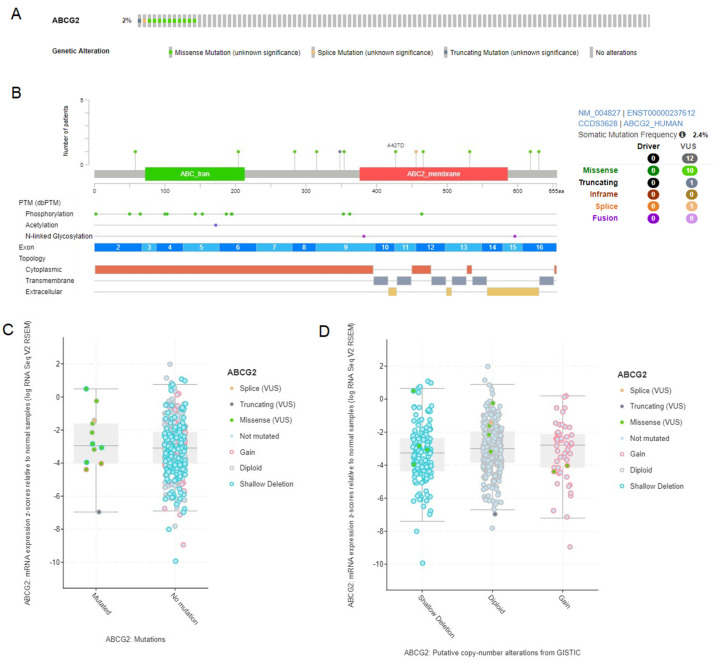
The graphical representation and analysis of mutations and copy number alterations in the *ABCG2* gene for LUAD samples, using cBioPortal. (**A**) The Oncoprint of overall genetic alterations. (**B**) The distribution of mutations and their effect on the protein, exon, and topology spans. (**C**) Association between mutations and mRNA expression z-scores relative to normal samples. (**D**) Association between copy number alterations and mRNA expression z-scores relative to normal samples.

**Figure 7 biomedicines-12-02394-f007:**
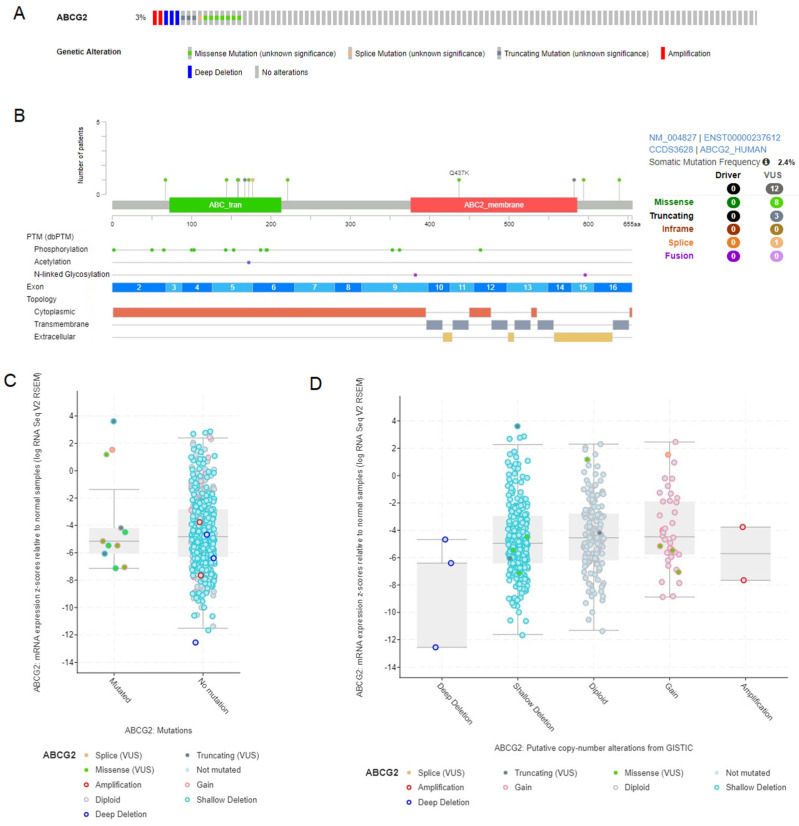
The graphical representation and analysis of mutations and copy number alterations in the ABCG2 gene for LUSC samples, using cBioPortal. (**A**) The Oncoprint of overall genetic alterations. (**B**) The distribution of mutations and their effect on the protein, exon, and topology spans. (**C**) Association between mutations and mRNA expression z-scores relative to normal samples. (**D**) Association between copy number alterations and mRNA expression z-scores relative to normal samples.

**Figure 8 biomedicines-12-02394-f008:**
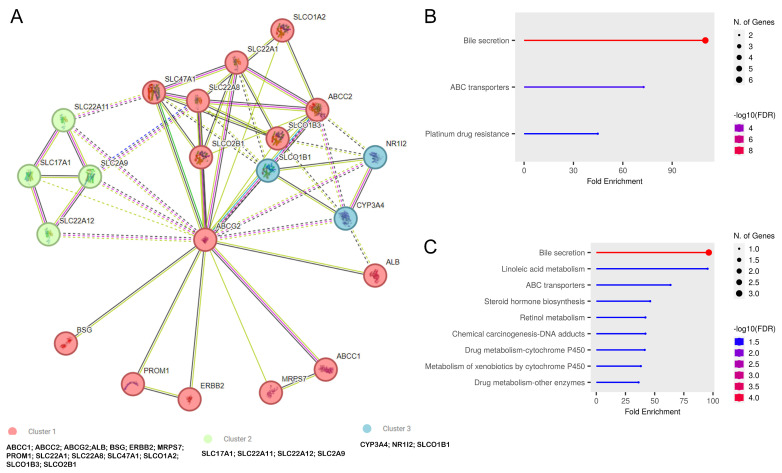
The protein–protein interaction (PPI) network and functional enrichment of ABCG2. (**A**) PPI networks of 20 interacting proteins correlated with ABCG2 using the STRING tool (dashed-lines represents the inter-cluster edges). (**B**,**C**) The analysis of pathways associated with gene clusters indicated in PPI networks: (**B**) cluster 1 and 2, (**C**) cluster 3 (ShinyGO 0.80).

**Table 1 biomedicines-12-02394-t001:** Characteristics of the investigated group.

Variable	No. of Patients
Age (range, years)	67 (32–82)
Gender:	
Male	37
Female	12
Tobacco smoking status:	
Non-smokers	22
smokers	27
Histological type of cancer:	
LUAD	19
LUSC	30
Cancer stage:	
I	22
II	16
III	11
IV	0
Grade of histological malignancy:	
G1	2
G2	35
G3	12
Chemotherapy:	
Yes	13
No	36

## Data Availability

In the case of the in silico analysis, data are derived from publicly available datasets (TIMER 2.0, UALCAN, TNMplot, MEXPRESS, cBioPortal, MethSurv, KM Plotter, STRING, ShinyGO 0.80). In the case of the wet lab, data are available on request from the Corresponding Author.

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
