# Peer review of "ABCG2 Gene Expression in Non-Small Cell Lung Cancer"

_biomedicines, 2024, doi:10.3390/biomedicines12102394_

Round 1
Reviewer 1 Report
Comments and Suggestions for Authors
This study investigated the role of ABCG2 (Breast Cancer Resistance Protein) in non-small cell lung cancer (NSCLC) development, prognosis, and patient outcomes using publicly available data and blood samples from 50 patients. The findings showed low ABCG2 expression in NSCLC, with no correlation to clinical aggressiveness, but higher expression improved overall survival in lung adenocarcinoma (LUAD) patients. The study highlighted CpG sites influencing prognosis and drug resistance. The results suggest that ABCG2 may be important for lung cancer patient survival and immune-related therapy outcomes, supporting personalized medicine approaches.
Below are my comments:
Introduction part provides a good overview of lung cancer, the role of ABCG2, and its impacts on cancer treatment and drug resistance. However, The introduction does not focus on the relevance of ABCG2 and NSCLC which is the title of this article. I would suggest the authors to cut down some irrelevant context and add more review on ABCG2 and NSCLC.
Research design and methodology are adequately described and relevant references have been provided.
The results are well and clearly described, but the image quality in most of the figures is very poor. The authors should improve the resolutions of the images.
The discussion section thoroughly addresses the results and provides adequately comparison to previous studies.
Author Response
October 5th 2024
Editor of BIOMEDICINES
Dear Editor,
ID biomedicines-3236257
Manuscript of the article entitled “ABCG2 gene expression in non-small cell lung cancer”
We appreciate the time and effort that you dedicated to providing feedback on our manuscript and are grateful for the valuable improvements to our paper. The manuscript has been revised according to the suggestions. Those changes are highlighted (in red) within the introduction section.
Comments 1: Introduction part provides a good overview of lung cancer, the role of ABCG2, and its impacts on cancer treatment and drug resistance. However, The introduction does not focus on the relevance of ABCG2 and NSCLC which is the title of this article. I would suggest the authors to cut down some irrelevant context and add more review on ABCG2 and NSCLC.
Response 1: A large part of the second paragraph was removed. In the paragraphs below, single sentences unrelated to topic of this study have been removed. As presented in the Introduction section, ABCG2 expression has been studied multiple times in non-small cell lung cancer, but mainly at the protein level and focusing on the influence of individual drugs used in studies on it concentration. A review of the available literature reveals relatively few studies where ABCG2 gene expression and its correlation with the NSCLC development, course of cancer disease and patient's prognosis is main objective of examination. These research are listed and commented in the Discussion section. The Introduction section also includes changes suggested by other Reviewers. We hope that all the changes introduced will be sufficient for this part of the manuscript to be positively received and accepted.
Now: Introduction
Lung cancer is the most common cancer among men and the second most common in women worldwide. This type of cancer is also leading cause of cancer-related deaths among over 30 groups of neoplasms in half of the regions analyzed worldwide (among both, men and women). While tobacco smoking remains the most important factor predisposing to development of the disease [1,2], up to a quarter of all lung cancer cases occur in lifelong never-smokers. The incidence of lung cancer is decreasing as smoking rates decline, however, the incidence of lung cancer in never-smokers remains stable or increase, and may depend on gender, age or ethnicity. In addition to the effects of smoking, the increasing attention is being paid to the indoor/workplace exposure to a radon, asbestos, arsenic, cooking oil or exhaust fumes and level of other pollutants in the air we live in. Other related factors are family lung cancer history, diet, infectious and inflammatory diseases [3,4].
Lung cancer, according to histological type, is divided into two broad categories: non-small cell lung cancer (NSCLC) and small cell lung cancer (SCLC). NSCLC which is the most commonly diagnosed histological type of lung cancer, is further classified into three subtypes: squamous-cell carcinoma, adenocarcinoma, and large-cell carcinoma. In smokers, SCLC, squamous cell carcinoma and large cell carcinoma predominate, whereas in never-smokers, adenocarcinoma is the most common. Tumors in smokers arise in tissue that has been genetically significantly altered by exposure to cigarette smoke, whereas in nonsmokers the number of mutations is lower, but each of them significantly contributes to malignant transformation. A characteristic molecular feature of adenocarcinomas in never-smokers is the presence of one of the common mutations of EGFR gene, rare KRAS mutation, or ALK or ROS1 rearrangements. A rare EGFR mutation, a common KRAS mutation and diverse alterations in TP53 gene are more frequently detected in smokers with NSCLC. In response to these observations, treatment schedules that include mutation-targeted therapy have been developed and are now widely used in clinical practice. The constant rate of non-smokers developing NSCLC highlights the important role of non-modifiable factors that are associated with carcinogenesis in lung tissue. Differences between smokers and non-smokers with NSCLC also concern other features such as gene expression patterns or germline polymorphisms [5-7]. CurrentlyMoreover, the major obstacles to the successful management of lung cancer are diagnosis at advanced or locally advanced stages, limited number of markers to predict the response to applied therapy and monitoring its efficacy, as well as development of resistance to therapy leading to nonresponse or insufficient response. Therefore, researchers in this field need to expand knowledge with next promising molecular alterations that may bring benefits in clinical practicein this study, we analyze a promising molecular factor that may be important for the development of NSCLC and bring benefits in clinical practice.
ABCG2 (ATP-binding cassette superfamily member G 2) is a memberThe members of the ATP-binding cassette (ABC) transporter superfamily that isare primarily known to significantly alter the pharmacokinetics of most anticancer drugs, including traditional chemotherapeutics, as well as molecularly targeted therapeutic agents. However, according to our previous study, the gene encoding the best known ABC transporter - ABCB1, may also be a promising indicator of carcinogenesis and a disease prognostic factor, which could contribute to alteration of cancer progression and survival of patients [8]. The second factor of similar importance may be ABCG2 (ATP-binding cassette superfamily member G 2).In certain cancers, oOverexpression of ABCG2 associated with drug efflux may be among mechanisms of multidrug resistance (MDR), leading to failure in cancer therapy. The efficacy of targeted inhibition of ABCG2-mediated transport has been investigated in various cancers such as e.g. colon cancer [89], breast cancer [910] and leukemia [1011]. After the first disappointing results, it was necessary to continue the search for new strategies, promising multipotential MDR modulators and further preclinical tests. In lung cancer, an association was found between the ABCG2 transporter and the frequency of osimertinib adverse events [1112], the risk of resistance to platinum therapy [1213], or adagrasib penetration of target tissues [1314]. New MDR modulators such as tazemetostat [1415], I-CBP112 - CBP/EP300 bromodomain inhibitor [1516] or sonidegib [1617] may increase the tumor cells' sensitivity to anti-cancer agents by functional inhibition of ABCG2. However, these new drugs could bring the most benefits in clinical management of NSCLC patients with high ABCG2 expression.
Under normal physiological conditions, members of the ABCG subfamily are responsible for the regulation of cholesterol and bile acids homeostasis, steroid hormones transport, heme metabolism and hypoxic signaling. Their presence at the strategic sites in the body along with the results of studies conducted on animals and in human cancer samples suggest that ABCG members play side population (SP) cells an important role in tumor formation and several other cancer-related processes. In addition, ABCG2 is a marker of the side population (SP) cells, representing pluripotent cancer stem cells. According to more recent data, the features of cancer that are significantly associated with activity of ABCG2 include genomic instability, resistance to cell death, plasticity of the cancer stem cell phenotype, sustaining proliferative signaling/ability to evade growth suppression, epigenetic reprogramming and polymorphic differences in the microbiomes [17,18,19].
ABCG2 seems to be an important element of defense against carcinogens and factors that can stimulate inflammation. This was proven in a mouse model where knockout of the gene analogue caused toxic reactions and carcinogen accumulation Knockout the Bcrp1 gene, which is the homologue of human ABCG2, in mice resulted in hypersensitivity to the dietary chlorophyll catabolites and contribute to several porphyrin-related phototoxicities [1920,21]. Bcrp1 limited the accumulation of 2-amino-1-methyl-6-phenylimidazo[4,5-b]pyridine (PhIP), a food-born chemical carcinogen, in mouse organs [20]. ABCG2 also has the ability to pump out some of the polycyclic aromatic hydrocarbons contained in tobacco smoke. Therefore, it protects both normal lung cells and cancer cells from the effects of carcinogenic toxicity. It is known that smoking during the cancer therapy worsens the patients' prognosis, possibly due to the fact that protection of cells from exposure to smoke involves similar mechanisms, as the development of resistance to anti-cancer drugs [2122].
One of the biological effects of prolonged exposure to carcinogens and/or substances that affect the immune system is production of reactive oxygen species (ROS), that may induce e.g. genomic instability, epigenetic modifications, immunomodulation or chronic inflammation, any of which can initiate carcinogenesis. The results obtained in human colorectal adenocarcinoma cell line indicated that ROS promoted inflammation, which was more intense when ABCG2 expression was decreased. It is likely that the downregulation of the ABCG2 contributes to the activation of NF-κB signaling pathway, which controls multiple aspects of the immune response [2223]. Low ABCG2 levels were observed in colon patients with active inflammation, colon adenomas and carcinomas compared to morphologically normal tissue as well as to healthy individuals. It may therefore be an indicator of the activation of processes leading to the development of cancer. At the same time, molecular changes in ABC transporters (including ABCG2) may have an effect on the transport of chemical compounds affecting the microbiome and/or produced by them [2324]. Similarly to colorectal cancer, inflammation in the respiratory tract may be preceded by altered lung microbial community composition and microbiome outside the lungs [2425]. The diet, environmental factors and ABC transporters mutually influence the complex network of interaction between the lung, oropharynx and gut microbiome. For example, a significantly increased risk of lung cancer has also been observed in patients with lower respiratory tract infections or H. pylori presence [25,26,27].
Cancer tissue is organized in hierarchical heterogeneous cell populations, that exhibit distinct phenotypes and function within the tumor. Among them of particular interest are cancer stem cells (CSCs) which are derived from the initiated normal stem/progenitor cells and have the ability to self-renewal and differentiation into multiple cell types. These properties enable tumorigenesis, metastasis and provide intra-tumor heterogeneity. Cancer SP is a small fraction of tumor cells enriched with CSCs, that can be isolated/sorted using appropriate cell functional assays and surface markers. ABCG2 plays an important role in the enhancement of CSCs tumorigenic potential. SP cells isolated from lung cancer cell lines demonstrated a higher level of ABCG2 at the mRNA and protein levels, providing them broad spectrum of drug/xenobiotics resistance [27,28,29]. The expression of ABCG2 which is responsible for phenotypic characteristics of SP cells, is also involved in the WNT/β-Catenin, Hedgehog, Hippo and PI3K/Akt/mTOR signaling pathway [2930-3132]. Therefore, ABCG2 status has a significant impact on tumor initiation and progression, the risk of local recurrence and metastasis to another part of the body, and patient survival.
This study involves the analysis of molecular and phenotypic datasets collected by The Cancer Genome Atlas (TCGA) and other bioinformatics databases. Using massive amounts of data from multiple studies and centers can yield more robust results and more reliable findings. Cancer bioinformatics plays an important role in the preliminary identification and/or authentication of promising biomarkers, which are revealed from individual clinical-cases research. In-depth study of ABCG2 gene expression becomes possible with selection and processing of transcriptomic and epigenomic data collected during multiomics studies in non-small cell lung cancer [33,34].
Here, we primarily investigate the impact of ABCG2 gene expression on NSCLC development, course of cancer disease and patient prognosis using data collected in databases. Analyses conducted using bioinformatics tools are additionally supplemented with the results of our own wet research. In this study, we also explore the metabolic pathways in which ABCG2 is involved and its interactions with other proteins. The obtained results were compared with the available literature and discussed thoroughly.
Comments 2: Research design and methodology are adequately described and relevant references have been provided.
Response 2: Thank you!
Comments 3: The results are well and clearly described, but the image quality in most of the figures is very poor. The authors should improve the resolutions of the images.
Response 3: The quality of most figures has been improved. The size of the figures was standardized. All figures have a resolution of 300 dpi and/or 600 dpi.
Comments 4: The discussion section thoroughly addresses the results and provides adequately comparison to previous studies.
Response 4: Thank you!
Reviewer 2 Report
Comments and Suggestions for Authors
The manuscript „ABCG2 gene expression in non-small cell lung cancer“ by Jeleń et al. aims to investigate the expression pattern of ABCG2 in NSCLC, focusing on lung adenocarcinoma (LUAD) and squamous cell carcinoma (LUSC) using publicly available datasets. The authors correlated ABCG2 expression with clinicopathological features and clinical outcome of lung cancer patients. The authors also checked methylation status of ABCG2, correlated expression of ABCG2 with tumor immune infiltration, analyzed gene mutation and copy number alteration and performed interactome analysis of ABCG2 using publicly available web tools.
The topic is very interesting and authors used wide range of web tools to perform their in silico analysis. Nevertheless, several major concerns should be addressed before considering the manuscript for publication.
The role of ABCG2 in LUAD has been extensively studied over the years and therefore in some aspects paper lacks novelty which is one of the major issues of the study. Thus, the authors need to emphasize what are crucial, novel findings of here presented study.
In Figure 4, authors showed correlation of ABCG2 expression and patient survival. The authors observed statistically significant correlation between increased ABCG2 expression and higher survival of patients with lung cancer (Fig 4A, 4D and 4G). However it is unclear whether lung cancer patients also includes patients with SCLC. Could authors please comment.
General remark also refers to some language misspellings/typos throughout the text, so I recommend the authors to carefully read and revise the manuscript accordingly. Editing of English language is required in the manuscript.
Throughout the text, there are many sentences that are not understandable and should be revised to make them more clear to reader. Some of the examples are:
Lines 88-92: “Their presence at strategic sites in the body along with the results of studies conducted on animals and in human cancer samples suggest that they play side population (SP) cells an important role in tumor formation and several other cancer-related processes. In addition, ABCG2 is a marker of, representing pluripotent cancer stem cells.”
Lines 113-114: “It is likely that downregulation of ABCG2 contributes to activation of NF-κB signaling pathway, which controlling multiple aspects of immune response [22].”
Lines 115-117: “Low ABCG2 levels were observed in colon patients with active inflammation, colon adenomas and carcinomas compared to morphologically normal tissue as well as to healthy individuals. It may therefore be an indicator of the activation of processes leading to the development of cancer.”
Some statements should be supported by references. For example:
Lines 389-390: “The expression of the ABCG2 transporter can be regulated by diverse signaling molecules derived from immune cells. On the other hand, cytokines are transported and controlled by various ABC family proteins.”
Comments on the Quality of English Language
Editing of English language is required
Author Response
October 6th 2024
Editor of BIOMEDICINES
Dear Editor,
ID biomedicines-3236257
Manuscript of the article entitled “ABCG2 gene expression in non-small cell lung cancer”
We appreciate the time and effort that you dedicated to providing feedback on our manuscript and are grateful for the valuable improvements to our paper. The manuscript has been revised according to the suggestions. Those changes are highlighted (in red) within the introduction section.
Comments 1: The role of ABCG2 in LUAD has been extensively studied over the years and therefore in some aspects paper lacks novelty which is one of the major issues of the study. Thus, the authors need to emphasize what are crucial, novel findings of here presented study.
Response 1: Thank you for poinitng this out. We agree that ABCG2 expression has been studied multiple times in non-small cell lung cancer, however primarily at the protein level. Most of these studies concern on the role of ABCG2 protein in drug resistance and the researches were conducted on cell lines. A review of the literature reveals relatively few studies where ABCG2 gene expression and its correlation with the course of cancer disease and patient's prognosis is main objective of examination. For NSCLC there are only few this type of studies available, which prevents any conclusions from being drawn. We have tried to point out in the Conslusions section. Following the reviewer's suggestion, this subsection has been expanded and reworded.
Now: Conclusions
This study demonstrates the potential of ABCG2 gene expression as well as its methylation evaluation as a biomarker in cancer through in-silico and wet analysis. The low ABCG2 mRNA level is a feature of NSCLC, with no correlation to clinical aggressiveness. This gene appears to have a particularly significant impact on the survival of patients because higher expression improved overall survival in LUAD patients. Additionally, the study highlighted CpG sites influencing prognosis and drug resistance. with lung cancer and on tThe effect of immunotherapy related to immune cell infiltration also significantly depends on the ABCG2 expression. To the best of our knowledge, this study is the first to present the most comprehensive analysis of expression of this gene in NSCLC. Despite the fact that, there were no differences of ABCG2 gene expression in blood between subgroups of investigated lung cancer patients (before surgery, 100 days and one year after the surgical removal of the tumor), it is the first such investigation, hence it was not possible to confront/discuss obtained results and draw unequivocal conclusions. Thereby, theseSome findings of this study require to confirmconfirmation oin a larger group of participants.
Comments 2: In Figure 4, authors showed correlation of ABCG2 expression and patient survival. The authors observed statistically significant correlation between increased ABCG2 expression and higher survival of patients with lung cancer (Fig 4A, 4D and 4G). However it is unclear whether lung cancer patients also includes patients with SCLC. Could authors please comment.
Response 2: Agree. The caption does not precisely indicate which group of lung cancer patients is shown in Figures 4A, 4D, and 4G. The analysis includes only NSCLC. This information was clarified in the description of the figure.
Now: Figure 4. Kaplan-Meier survival curves of ABCG2 (low vs. high expression level) overallin (A, D, G) in non-small cell lung cancer, (B, E, H) in LUAD and (C, F, I) LUSC (p < 0.05; Kaplan–Meier ploter). OS, overall survival; FP, first-progression survival; PPS, post-progression survival; HR, hazard ratio; red: high expression; black: low expression.
Comments 3: General remark also refers to some language misspellings/typos throughout the text, so I recommend the authors to carefully read and revise the manuscript accordingly. Editing of English language is required in the manuscript.
Response 3: Language correction was done.
Comments 4: Throughout the text, there are many sentences that are not understandable and should be revised to make them more clear to reader. Some of the examples are:
Response 4:
Before: Lines 88-92: “Their presence at strategic sites in the body along with the results of studies conducted on animals and in human cancer samples suggest that they play side population (SP) cells an important role in tumor formation and several other cancer-related processes. In addition, ABCG2 is a marker of, representing pluripotent cancer stem cells.”
Now: Their presence at the strategic sites in the body along with the results of studies conducted on animals and in human cancer samples suggest that ABCG members play an important role in tumor formation and several other cancer-related processes. In addition, ABCG2 is a marker of the side population (SP) cells, representing pluripotent cancer stem cells.
Before: Lines 113-114: “It is likely that downregulation of ABCG2 contributes to activation of NF-κB signaling pathway, which controlling multiple aspects of immune response [22].”
Now: It is likely that the downregulation of the ABCG2 contributes to the activation of NF-κB signaling pathway, which controls multiple aspects of the immune response [23].
Before: Lines 115-117: “Low ABCG2 levels were observed in colon patients with active inflammation, colon adenomas and carcinomas compared to morphologically normal tissue as well as to healthy individuals. It may therefore be an indicator of the activation of processes leading to the development of cancer.”
Now:
This sentence was removed due to the other Reviewers’ suggestion.
Comments 5: Some statements should be supported by references. For example:
Lines 389-390: “The expression of the ABCG2 transporter can be regulated by diverse signaling molecules derived from immune cells. On the other hand, cytokines are transported and controlled by various ABC family proteins.”
Response 5: The above statement was supported by two new citations [49 and 50]. The citations were added in the appropriate place in the References section.
[49] Lu, X.; Chen, M.; Shen, J.; Xu, Y.; Wu, H. IL-1β functionally attenuates ABCG2 and PDZK1 expression in HK-2 cells partially through NF-ĸB activation. Cell Biol Int. 2019, 43, 279-289. https://doi.org/10.1002/cbin.11100
[50] Mosaffa, F.; Kalalinia, F.; Lage, H.; Afshari, J.T.; Behravan, J. Pro-inflammatory cytokines interleukin-1 beta, interleukin 6, and tumor necrosis factor-alpha alter the expression and function of ABCG2 in cervix and gastric cancer cells. Mol Cell Biochem. 2012, 363, 385-393. https://doi.org/10.1007/s11010-011-1191-9
Reviewer 3 Report
Comments and Suggestions for Authors
Journal of Biomedicines
Research Article;
The article entitled “ABCG2 gene expression in non-small cell lung cancer’’. The scientists tried to examine the ATP-binding cassette subfamily G member 2 (ABCG2), which plays a role in multidrug resistance processes and serves as a marker for side population cells in human malignancies. The ABCG2 gene did not correlate with the clinical aggressiveness of lung cancer. Elevated ABCG2 expression enhanced overall survival, but only in LUAD. CpG sites situated on the CpG island that affect the prognosis of NSCLC patients were identified. The laboratory findings indicated no changes in ABCG2 expression levels in blood samples obtained from patients at various periods during the diagnostic-therapeutic process. The outcomes of this research may endorse customized medicine approaches grounded on bioinformatics results.
I meticulously reviewed the work and deemed it appropriate for publishing in the journal. I consent to the potential publishing of this work. The article has many prevalent errors that need correction by the writers. Upon rectifying all errors, the manuscript may be deemed suitable for publishing in the esteemed Biomedicines Journal.
Comments for Authors
Ø Section Introduction: The author needs to include about one paragraph of bioinformatics important in the introduction section on diseases and cancer.
Ø Why did the author select the ABCG2 gene, there are several genes downregulated or upregulated during cancer, but why did the author directly target the ABCG2 gene?
Ø It would be better for the author to perform the immunostaining and western blotting to confirm in vivo the effect of ABCG2 in lung cancer.
Ø As the author didn’t discuss the target region of ABCG2, the correlation study of the main upregulated genes with ABCG2 and the author could include the main angiogenesis or metastasis pathway-related genes or apoptosis or ferroptosis-related genes could be targeted by ABCG2.
Ø The author tried wonderfully and may the study help him to increase the impact of the manuscript by following and citing the article DOI: 10.2174/0113892037269589231017055642, DOI: 10.2174/0113892037269589231017055642
Ø The image size and expression in the manuscript are not standard. The needs to paste full-size images 300 or 600 dpi. And also rearrange the figure size.
Ø Use EndNote or Mendeley software for reference sequences.
Ø Check grammar and spelling throughout the manuscript. There are some mistakes.
Author Response
October 7th 2024
Editor of BIOMEDICINES
Dear Editor,
ID biomedicines-3236257
Manuscript of the article entitled “ABCG2 gene expression in non-small cell lung cancer”
We thank the Reviewer for the detailed comments and suggestions. We believe that the manuscript has been substantially improved. The manuscript was edited for grammar. Our incorporation of the Reviewers’ suggestions is as follows:
- all changes were marked in red
Comments 1: Section Introduction: The author needs to include about one paragraph of bioinformatics important in the introduction section on diseases and cancer.
Response 1: A new paragraph describing the application of bioinformatics data in cancer research was added to the Introduction section. The additional citations were added in the appropriate place in the References section.
This study involves the analysis of molecular and phenotypic datasets collected by The Cancer Genome Atlas (TCGA) and other bioinformatics databases. Using massive amounts of data from multiple studies and centers can yield more robust results and more reliable findings. Cancer bioinformatics plays an important role in the preliminary identification and/or authentication of promising biomarkers, which are revealed from individual clinical-cases research. In-depth study of ABCG2 gene expression becomes possible with selection and processing of transcriptomic and epigenomic data collected during multiomics studies in non-small cell lung cancer [33,34].
[33] Nelakurthi, V.M.; Paul, P.; Reche, A. Bioinformatics in Early Cancer Detection. Cureus. 2023, 15, e46931. https://doi.org/10.7759%2Fcureus.46931
[34] Jiang, P.; Sinha, S.; Aldape, K.; Hannenhalli, S.; Sahinalp, C.; Ruppin, E. Big data in basic and translational cancer research. Nat Rev Cancer. 2022, 22, 625-639. https://doi.org/10.1038/s41568-022-00502-0
Comments 2: Why did the author select the ABCG2 gene, there are several genes downregulated or upregulated during cancer, but why did the author directly target the ABCG2 gene?
Response 2: The ABC gene superfamily is crucial for growth, development, and responses to environmental stresses in most cells of our body. First, we examined the ABCB1 gene and checked whether its expression is important in non-small cell lung cancer. Promising research results prompted us to analyze another, equally important factor belonging to this family, ABCG2. A short explanation was added to the Introduction section. The one new citation was added in the appropriate place in the References section.
Now: ABCG2 (ATP-binding cassette superfamily member G 2) is a memberThe members of the ATP-binding cassette (ABC) transporter superfamily that isare primarily known to significantly alter the pharmacokinetics of most anticancer drugs, including traditional chemotherapeutics, as well as molecularly targeted therapeutic agents. However, according to our previous study, the gene encoding the best known ABC transporter - ABCB1, may also be a promising indicator of carcinogenesis and a disease prognostic factor, which could contribute to alteration of cancer progression and survival of patients [8]. The second factor of similar importance may be ABCG2 (ATP-binding cassette superfamily member G 2). (…)
[8] Jeleń, A.M.; Strehl, B.; Szmajda-Krygier, D.; Pązik, M.; Balcerczak, E. Bioinformatics-Based Characterization of ATP-Binding Cassette Subfamily B Member 1 (ABCB1) Gene Expression in Non-Small-Cell Lung Cancer (NSCLC). Appl. Sci. 2023, 13, 6576. https://doi.org/10.3390/app13116576
Comments 3: It would be better for the author to perform the immunostaining and western blotting to confirm in vivo the effect of ABCG2 in lung cancer.
Response 3: We decided not to perform the immunostaining and western blotting becouse our study concerned mRNA, not protein. There are many research on this transporter at the protein level. Only limited number of studies evaluate this factor at the transcript level. The main goal of our study was to capture important associations that would explain what changes the ABCG2 transcript undergoes, even before the protein formation.
Comments 4: As the author didn’t discuss the target region of ABCG2, the correlation study of the main upregulated genes with ABCG2 and the author could include the main angiogenesis or metastasis pathway-related genes or apoptosis or ferroptosis-related genes could be targeted by ABCG2.
Response 4: At the end of the Discussion section, two new paragraphs were added discussing selected genes related to angiogenesis or metastasis pathway or genes related to apoptosis or ferroptosis. The additional citations were added at the end of the References section.
Now: The ABCG2 gene may also be involved in the process of carcinogenesis indirectly, by influencing such processes as apoptosis, autophagy or ferroptosis. In a normal placenta, reduced or no expression of this gene resulted in increased sensitivity of placental trophoblast cells to apoptosis in response to cytokines or ceramides. Therefore, the presence of ABCG2 gene expression protects the cell from apoptosis. However, in the case of cancer cells, this may have a negative effect, leading to excessive proliferation of these cells and promoting carcinogenesis. Additionally, it has been shown that overexpression of ABCG2 in cancer cells enhances the process of autophagy induced by non-substrate stressors (such as radiation or nutrient starvation), contributing to increased cell survival and promoting cancer progression [72,73]. The ABCG2 as a potential heme exporter may be associated with the effect on ferroptosis, although the mechanism has not been fully elucidated, but it is believed that the amount of heme in the cell affects their sensitivity to this phenomenon. Reducing the amount of heme potentially lowers ferroptosis [74].
Although it has been shown in a mouse model that vascular endothelial progenitor cells with overexpression of ABCG2 have a higher potential for vessel formation in vivo compared to mature endothelial cells, no association has been demonstrated between ABCG2 gene expression and VEGF gene expression and tumor vascularization in patients with retionoblastoma [75,76].
[72] Evseenko, D.A.; Murthi, P.; Paxton, J.W.; Reid, G.; Emerald, B.S.; Mohankumar, K.M.; Lobie, P.E.; Brennecke, S.P.; Kalionis, B.; Keelan, J.A. The ABC transporter BCRP/ABCG2 is a placental survival factor, and its expression is reduced in idiopathic human fetal growth restriction. FASEB J. 2007, 21, 3592-605. doi: 10.1096/fj.07-8688com
[73] Ding, R.; Jin, S.; Pabon, K.; Scotto, K.W. A role for ABCG2 beyond drug transport: Regulation of autophagy. Autophagy. 2016, 12, 737-751. doi: 10.1080/15548627.2016.1155009
[74] Wang, Y.; Zhang, Z.; Jiao, W.; Wang, Y.; Wang, X.; Zhao, Y.; Fan, X.; Tian, L.; Li, X.; Mi, J. Ferroptosis and its role in skeletal muscle diseases. Front Mol Biosci. 2022, 9, 1051866. doi: 10.3389/fmolb.2022.1051866
[75] Kim, M.; Kim, J.H.; Kim, J.H.; Kim, D.H.; Yu, Y.S. Differential expression of stem cell markers and vascular endothelial growth factor in human retinoblastoma tissue. Korean J Ophthalmol. 2010, 24, 35-9. https://doi.org/10.3341/kjo.2010.24.1.35
[76] Lin Y, Gil CH, Banno K, Yokoyama M, Wingo M, Go E, Prasain N, Liu Y, Hato T, Naito H, Wakabayashi T, Sominskaia M, Gao M, Chen K, Geng F, Gomez Salinero JM, Chen S, Shelley WC, Yoshimoto M, Li Calzi S, Murphy MP, Horie K, Grant MB, Schreiner R, Redmond D, Basile DP, Rafii S, Yoder MC. ABCG2-Expressing Clonal Repopulating Endothelial Cells Serve to Form and Maintain Blood Vessels. Circulation. 2024, 150, 451-465. https://doi.org/10.1161/circulationaha.122.061833
Comments 5: The author tried wonderfully and may the study help him to increase the impact of the manuscript by following and citing the article DOI: 10.2174/0113892037269589231017055642, DOI: 10.2174/0113892037269589231017055642
Response 5: A new paragraph was added in the Discussion section, which quoted the above article. New citations were added in the References section.
Now: Infiltration of immune cells, particularly macrophages, may be the result of EPHB2 overexpression which is characteristic for LUAD. Eph receptor B2 (EphB2), an important member of the ephrin receptor family, can stimulate ABCG2 expression in other types of cancer cells, which can explain the correlation observed in our study [65]. In the case of other members of this superfamily, their overexpression has also been observed during the development of lung cancer. An example is EphB4, which has been linked to tumors angiogenesis, growth, and metastasis [66]. Assessment of the interaction between Eph receptors and ABCG2 requires further research.
[65] Liu, W.; Yu, C.; Li, J.; Fang, J. The Roles of EphB2 in Cancer. Front Cell Dev Biol. 2022, 10, 788587. doi: 10.3389/fcell.2022.788587
[66] Ullah, A.; Razzaq, A.; Zhou, C.; Ullah, N.; Shehzadi, S.; Aziz, T.; Alfaifi, M.Y.; Elbehairi, S.E.I.; Iqbal, H. Biological Significance of EphB4 Expression in Cancer. Curr Protein Pept Sci. 2024, 25, 244-255. doi: 10.2174/0113892037269589231017055642
Comments 6: The image size and expression in the manuscript are not standard. The needs to paste full-size images 300 or 600 dpi. And also rearrange the figure size.
Response 6: The size of the figures was standardized. All figures have a resolution of 300 dpi and/or 600 dpi.
Comments 7: Use EndNote or Mendeley software for reference sequences.
Response 7: The Journal 'recommend' preparing the references with a bibliography software package, such as EndNote or another software. The use of these softwares is not strictly required. Bibliographic descriptions were prepared in accordance with the ACS style. Additionally, a DOI number was added wherever possible.
Comments 8: Check grammar and spelling throughout the manuscript. There are some mistakes.
Response 8: Language correction was done.
Round 2
Reviewer 2 Report
Comments and Suggestions for Authors
I thank the authors for improving their manuscript which, to my opinion, in the revised form meets the standard of Biomedicines.